Enhancing machine learning-based forecasting of chronic renal disease with explainable AI

Singamsetty Sanjana 1
Ghanta Swetha 1
Biswas Sujit sujit.biswas@northumbria.ac.uk 2 3
Pradhan Ashok ashokkumar.p@srmap.edu.in 1
1 Department of Computer Science and Engineering, School of Engineering and Sciences, SRM University, AP , Guntur , Andhra Pradesh , India
2 Computer Science Department, Northumbria University , Newcastle , United Kingdom
3 Computer Science, University of London , London , United Kingdom
Wan Shibiao
Electronic publication date: 2024 Sep 26
Publication date: 2024
Volume: 10
Electronic Location ID: e2291
Received 2024 May 17; Accepted 2024 Aug 7
Copyright: ©2024 Singamsetty et al.
Copyright year: 2024
Copyright holder: Singamsetty et al.
License: This is an open access article distributed under the terms of the Creative Commons Attribution License, which permits unrestricted use, distribution, reproduction and adaptation in any medium and for any purpose provided that it is properly attributed. For attribution, the original author(s), title, publication source (PeerJ Computer Science) and either DOI or URL of the article must be cited.
License URL: https://creativecommons.org/licenses/by/4.0/

Keywords: Chronic renal disease, Machine learning, Explainable AI, GridSearchCV, Chronic kidney disease

Funding: The authors received no funding for this work.

==============================
Chronic renal disease (CRD) is a significant concern in the field of healthcare, highlighting the crucial need of early and accurate prediction in order to provide prompt treatments and enhance patient outcomes. This article presents an end-to-end predictive model for the binary classification of CRD in healthcare, addressing the crucial need for early and accurate predictions to enhance patient outcomes. Through hyperparameter optimization using GridSearchCV, we significantly improve model performance. Leveraging a range of machine learning (ML) techniques, our approach achieves a high predictive accuracy of 99.07% for random forest, extra trees classifier, logistic regression with L2 penalty, and artificial neural networks (ANN). Through rigorous evaluation, the logistic regression with L2 penalty emerges as the top performer, demonstrating consistent performance. Moreover, integration of Explainable Artificial Intelligence (XAI) techniques, such as Local Interpretable Model-agnostic Explanations (LIME) and SHapley Additive exPlanations (SHAP), enhances interpretability and reveals insights into model decision-making. By emphasizing an end-to-end model development process, from data collection to deployment, our system enables real-time predictions and informed healthcare decisions. This comprehensive approach underscores the potential of predictive modeling in healthcare to optimize clinical decision-making and improve patient care outcomes.

Introduction

Chronic renal disease (CRD) presents a formidable challenge in healthcare, characterized by compromised renal function affecting waste filtration and electrolyte balance. Its societal impact extends across individuals, families, and healthcare systems. Globally, CRD affects 10.4% of males and 11.8% of females, according to Chen, Zhang & Zhang (2016). Additionally, 13.3 million individuals experience acute kidney injury (AKI) annually, which can potentially culminate in CRD or kidney failure (Charleonnan et al., 2016). Consequently, the financial burden associated with CRD, encompassing diagnosis, treatment, and lifestyle adaptations, is considerable.

As CRD progresses, kidney function deteriorates, necessitating timely interventions to mitigate damage. This disease also places significant demands on healthcare systems, leading to decreased productivity and increased healthcare costs. Given the rising prevalence of CRD globally, the timely identification of individuals at risk and accurate prediction of disease progression are crucial for effective management and improved patient outcomes. The advent of machine learning (ML) methods has opened new avenues for analyzing complex medical data and aiding clinical decision-making, offering promise for the future of CRD management. However, concerns regarding the interpretability and reliability of ML models, often referred to as “black boxes”, have prompted the integration of Explainable AI (XAI) approaches to enhance transparency (Devam, Het Naik & Patel, 2023).

In response to the above challenges, our study aims to bridge the gap between advanced ML techniques and clinical practice by leveraging XAI methods to improve interpretability and transparency in CRD prediction. Our overarching goal is to empower healthcare professionals with data-driven insights while ensuring transparency and trust in machine-generated decisions. XAI techniques like Local Interpretable Model-agnostic Explanations (LIME) and SHapley Additive exPlanations (SHAP) provide actionable insights into model decisions and feature contributions facilitating informed medical decisions.

In this article, we investigate the application of ML models in predicting CRD onset and progression, with a focus on the utility of XAI techniques in supporting medical decision making. By employing SHAP and LIME, we aim to provide interpretable insights into the key features influencing model predictions enabling personalized interventions. Our study encountered several challenges such as missing values, irregular data ranges, and class imbalance. To address these, we employed strategies like missing value imputation, data scaling, and Synthetic Minority Over-sampling Technique (SMOTE) to mitigate class imbalance respectively. Additionally, we optimized hyperparameters using GridSearchCV to enhance model performance.

Several machine learning techniques are examined, including random forest, extra trees classifier, light gradient boosting, decision tree, logistic regression with L2 penalty, and artificial neural networks (ANN). After rigorous testing, random forest, extra trees classifier, logistic regression with L2 penalty, and ANN achieved the highest accuracy of 99.07%. The models were evaluated using various other metrics including recall, precision, F1-score, and AUC to ensure robustness and reliability. We further used SHAP and LIME to improve model explainability and interpretability.

Furthermore, for real-time chronic renal disease prediction, we developed a robust ML framework along with a fully functional app using Hugging Face. The app is publicly available at: https://huggingface.co/spaces/sanjana04/Chronic-Renal-disease-Prediction. This app offers patients or medical practitioners a user interface where they can find all the attributes contributing to CRD. The same attributes used in the dataset for model building are included in the application. Details about the dataset and the list of attributes are included in the Methodology section. Users input their values or existing disease status for each attribute and click on “predict”. The output returned is the class label “Kidney disease” (class label 1) or “No Kidney disease” (class label 0) based on the provided data. Figures 1 and 2 show the end-to-end application for CRD prediction. This application easily incorporates the high-performing logistic regression model, which has been upgraded with XAI approaches to deliver real-time predictions and actionable insights to healthcare practitioners. We demonstrate the practical application of advanced machine learning algorithms in clinical contexts by visualizing the model’s implementation sequence, facilitating informed decision-making and tuned patient care. The model’s implementation sequence is depicted in the Fig. 3, and the code is publicly available in our GitHub repository at https://github.com/sanjana-singamsetty/kidney_disease_prediction.

Figure 1 User interface for kidney disease predictor.

Figure 2 Sample CRD positive prediction from kidney disease predictor.

Figure 3 Proposed method for CRD prediction.

Related Work

The article builds on CKD with a particular emphasis on the application of machine learning (ML) and deep learning (DL) techniques to advance predictive modeling and risk assessment. These computational approaches encompass a wide range of ML algorithms, including support vector machines (SVM), decision trees (DT), random forest (RF), logistic regression, naive Bayes (NB) and k-nearest neighbors (KNN) (Sobrinho et al., 2020). The authors Ganie et al. (2023) also considered XGBoost and other ensemble-boosting methods for their study. In addition to machine learning models, extensive research has been conducted employing diverse deep learning models, including artificial neural networks (ANN) (Chittora et al., 2021), convolutional neural networks (CNN) (Mizdrak et al., 2022), and recurrent neural network architectures like long short-term memory (LSTM) networks, gated recurrent units (GRU), bidirectional LSTM, and bidirectional GRU (Akter et al., 2021), respectively.

Authors in this field employ a variety of assessment metrics, such as accuracy, precision, recall, F1-score, area under the receiver operating characteristic curve (AUC-ROC), and mean absolute error (MAE) for unsupervised feature selection (UFS) and model evaluation. By combining these computational techniques and rigorously assessing their performance, researchers aim to enhance and strengthen prediction models for CKD (Khan et al., 2020). The authors in (Debal & Sitote, 2022) utilized RF, SVM, DT, and feature selection methodologies to achieve an impressive 98.5% accuracy in CKD prediction. The authors in Alsekait et al. (2023) utilized stacking ensemble DL models, which include LSTM, CNN, GRU, and SVM, along with feature selection techniques to predict CKD. They integrated clinically confirmed data with AI-selected features to improve the prediction accuracy. The authors in Arif, Mukheimer & Asif (2023) introduce an ML model for predicting chronic kidney disease (CKD) enhancing clinical decision-making by integrating preprocessing steps, feature selection, and ML algorithms. Their research highlights the ML potential to improve clinical support systems and reduce the uncertainty in chronic disorder prognosis. Table 1 summarises the varied variety of ML and DL approaches used by authors in their work.

Table 1 Comparison of different approaches.

Author	Approach	Additional description	Accuracy	
Chittora et al. (2021)	Leveraged both ML and DL techniques, with the ANN achieving the highest accuracy.	Initially did not consider missing values and outliers.	99.6%	
Polat, Danaei Mehr & Cetin (2017)	KNN and SVM had been employed and the SVM algorithm, utilizing the FilterSubsetEval method with the Best First search engine, produced the best results.	A total of 400 cases are included, with 150 cases not having CRD and 250 cases having CRD.	98.5%	
Debal & Sitote (2022)	Three ML models, namely random forest, SVM, decision trees, were used along with two feature selection methods including Recursive Feature Elimination with Cross-Validated (RFECV), and Unsupervised Feature Selection (UFS). SVM and DT are their best performing models	The dataset they used consists of chronic kidney disease patient records between 2018 and 2019.	99.8%	
Wang, Chakraborty & Chakraborty (2020)	They have considered random forest, XG Boost, and Residual Network (RESNET) in ML approaches. They compiled their data by averaging the output from eight predictors to get better outcomes.	They used publicly available data from the National Health Insurance Sharing Service for this investigation.	97%	
Akter et al. (2021)	ANN, long short-term memory (LSTM), bidirectional LSTM, gated recurrent units (GRU), bidirectional GRU, multi-layer perceptrons (MLP), and simple recurrent neural networks (Simple RNN) have been used.	To combat overfitting, they have employed three feature selection techniques: filters, wrapper methods, and embedding methods.	99%	
Dritsas & Trigka (2022)	Naive Bayes, SVM, logistic regression, ANN, KNN, logistic model tree, random tree, random forest, J48, and rotation forest were employed in this investigation. AdaBoostM1, stochastic gradient descent, ensemble learning, and SMOTE were also applied to the provided data, and rotation forest yielded the highest results.	There are 400 cases in the dataset which they utilized in the research project, with 13 input characteristics and one target class.	99.2%	
Sobrinho et al. (2020)	J48 decision tree, random forest (RF), naive Bayes (NB), SVM, MLP, and KNN were among the machine learning techniques used in the study.	The study presented limitations such as a small dataset size.	95%	
Mizdrak et al. (2022)	Support vector machine (SVM) classifiers with local binary patterns and resilient speed-up features are used in the study, also using a convolutional neural network (CNN) model for raw pictures, deep features are obtained.	The article presents the Adaptive Hybridised Deep Convolutional Neural Network (AHDCNN) for predictive diagnosis of chronic renal illness.	97.33%	
Qin et al. (2019)	The approach utilizes the ML models like random forest and logistic regression to diagnose chronic kidney disease, showcasing the potential for accurate early detection.	For missing values, KNN imputation was used.	99.75%	
Khan et al. (2020)	The approach involves employing seven machine learning techniques, including NBTree, J48, and CHIRP, to classify CRD in a patient dataset.	Several evaluation metrics are used, like mean absolute error (MAE), root mean squared error (RMSE), recall, precision, and F-measure.	99.75%	

Methodology

Data preprocessing

Dataset

In our work, we utilize the CRD dataset, readily available from the open data repository curated by the University of California, Irvine (UCI) (Rubini & Eswaran, 2015). There are 400 instances in this dataset, 24 attributes, and 1 target class attribute. The first attribute is an ID and has been removed from the dataset since it is not required for the model. The other 23 features include demographic and medical measurements like age, blood pressure, specific gravity, albumin, sugar, red blood cells, pus cells, bacteria, blood glucose, blood urea, serum creatinine, sodium, potassium, haemoglobin, packed cell volume, white blood cell count, red blood cell count, hypertension, diabetes mellitus, coronary artery disease, appetite, pedal edema, and anaemia. Each feature’s type varies, including integer, continuous, categorical, and binary, with some missing values. The target variable, ‘class’ indicates the presence of CRD and contains no missing values. The dataset contains 250 cases labeled as ‘yes’ (indicating CRD) and 150 instances labeled as ‘no’ (indicating no CRD). The existence of missing values, a prevalent issue in real-world datasets, poses a serious barrier for this dataset. Random imputation techniques are employed to address this issue. It should be noted that out of the 400 instances in the dataset, only 158 instances contain no missing values. Figure 4 illustrates the attribute correlation heatmap representation, using Pearson correlation to provide insights into the relationships between various attributes.

Figure 4 Correlation matrix of data.

The characteristics in this dataset are classified into two types: numerical features and categorical features. Appetite, pus cell clumps, pus cells, diabetes mellitus, pedal edema, bacteria, red blood cells, hypertension, coronary artery disease, and anaemia are among the categorical characteristics. These characteristics, combined with the class attribute, determine the presence of CRD. Conversely, the remaining characteristics are categorized as numerical attributes. The class attribures are specifically used to determine whether an individual has CRD or not.

Missing value imputation

A random value imputation technique is used to address missing values for specific variables such as “anaemia”, “appetite”, “bacteria”, “coronary_artery_disease”, “diabetes_mellitus”, “red_blood_cells”, “pus_cells”, “peda_edema”, “pus_cell_clumps”, and “hypertension”. The requirement to manage missing data in these categorical variables motivates the usage of “random value imputation” in the CRD prediction study. Here, for a feature X with missing values, let N be the total number of missing values and Xobserved be the set of observed (non-missing) values in feature X. Now, randomly sample N values from Xobserved, resulting in N random values, denoted as Ri, where i ranges from 1 to N. Assign these random values to the corresponding missing values in feature X. (1) Xmissing=RandomSampleXobserved,N.

In summary, the random value imputation method is essential for dealing with missing data. It helps ensure that the CRD prediction model is built using as much complete and representative data as possible, particularly for categorical characteristics that play a role in the analysis.

_______________________ Algorithm 1 Proposed Method for CRD Prediction__________________________________ Input: CRD dataset Output: Predicted class label Begin Step 1:  Load the Dataset - Load the CRD dataset into a variable, such as “crd_data”. Step  2:  Handle  Missing  Values  -  Locate  and  handle  missing  data  in  the “crd_data” field. Step 3: Textual Data Encoding - Convert category text data in “crd_data” to numerical format. Step 4: Split and Scale the Data - Split the “crd_data” variable into subgroups for training and testing. If required, scale the data. Step 5: Hyperparameter tuning - Adjust the machine learning models’ hyper- parameters. Step 6: Apply ML Models - Train and test the machine learning models. Step 7: Determine Validation Scores - Evaluate model performance using val- idation scores. Step 8:  Interpreting ML models - Use SHAP and LIME explainability and interpretability to understand model decisions. End_____________________________________________________________________________________

Feature encoding

Once the missing values have been addressed, the dataset undergoes a transformation process where textual data is converted into a numerical format. This transformation is achieved using a feature encoding method called as“LabelEncoder”.

The LabelEncoder method serves the following purposes:

• Conversion of categorical to numeric: It is utilized to change categorical features, which originally contained text-based categories into a numerical representation. Each category received a distinct numeric code.

• Preservation of ordinal information: LabelEncoder maintains the relative order of the categories. This ensures that the assigned numeric codes reflect the original order or ranking of the categories. For instance, if the categories have a meaningful order like “low”, “medium”, and “high”, LabelEncoder retains this order when converting them into corresponding numeric codes (e.g., 0, 1, and 2).

The specific features that undergo this transformation include “pus_cell_clumps”, “red_blood_cells”, and “pus_cell”, which are transformed into two categories. Additionally, “anaemia”, “appetite”, “bacteria”, “diabetes_mellitus”, “hypertension”, “coronary_artery_disease”, and “peda_edema” are also transformed. The utilization of the LabelEncoder technique is crucial for converting this specific kind of data into numerical representations. Regarding studies on CRD, this improvement facilitates further data analysis, modeling, and forecasting operations. A numerical representation is being established for each of these attributes, simplifying data processing and enabling their application to a diverse variety of data-driven activities.

Splitting and scaling the data

Out of the 400 instances, the models are trained on seventy percent of the dataset (280 instances) and their performance is assessed on the remaining thirty percent (120 instances). In the training set, there are 180 ‘yes’ cases and 100 ‘no’ cases, while in the testing set, there are 70 ‘yes’ cases and 50 ‘no’ cases. The Standardisation and min-max scaling are two essential techniques that ensure the data features are centered around their respective means and maintain consistent scales. To create reliable and accurate projections, it is essential to engage in extensive information preparation. The MinMax Scaler is a data transformation technique that scales attributes to a specified range, often between 0 and 1. It maintains the arrangement of the initial distribution while modifying the data to a certain extent. (2) A=a−bc−b

where: A - is the scaled/normalized value of the feature. a - is the feature’s initial value. b - is the smallest score of the feature in the dataset. c - is the feature with the highest score in the dataset.

Using this formula, MinMax Scaler rescales each data point within the range [0, 1] depending on the dataset’s minimum and maximum values. It is frequently used to standardize characteristics and scale them to make them appropriate for machine learning techniques.

Standard scalar: StandardScaler follows the Standard Normal Distribution (SND), adjusting the data to have a unit variance and setting the mean to 0. This transformation is crucial for various machine learning algorithms as it helps to achieve better convergence and performance. (3) Standardization :Z=x−μσ.

Here, x represents the original data values, μ denotes the mean, and σ represents the standard deviation. The Standardization equation transforms the data values (x) into the standardized form (Z) with a mean of 0 and a standard deviation of 1.

Handling class imbalance

Within the course of our study on CRD prediction, we discovered a significant class imbalance in which CRD patients are significantly underrepresented relative to non-CRD cases. Recognizing the possible negative impact of this imbalance on our models’ predictive efficiency, we used the SMOTE, Synthetic Minority Over-sampling Technique proposed by Chawla et al. (2011) as a resampling approach, which is commonly utilised in medical contexts. By synthetically boosting instances of the minority class, SMOTE effectively alleviated the class imbalance, enabling a more balanced distribution for training purposes. This accurate method is designed to prevent the models from showing a bias towards the majority group, ensuring that they are capable of recognizing patterns across both CRD-positive and CRD-negative situations. By assuring a balanced representation of positive and negative instances, this method improves model predictability. Figure 5 illustrates how the data imbalance has been handled in the current process, resulting in more robust and unbiased predictions. After applying SMOTE to the training data, the class distribution was adjusted to correct class imbalances. Initially, the training data consisted of 280 instances, with 180 labelled as ‘yes’ (showing the presence of CKD) and 100 labelled as ‘no’ (indicating the absence of CKD). Following SMOTE, the data was resampled to ensure an equal number of cases in both classes. The training dataset now has 180 cases labelled ‘yes’ and 180 instances labelled ‘no’, resulting in a balanced dataset of 360 occurrences.

Figure 5 Employment of SMOTE technique for balancing the data.

Hyperparameter-optimized classifier comparison

The framework covers machine learning models, including extra trees classifier, random forest, LGBM, decision tree as well as other classification approaches like logistic regression. Furthermore, neural network models such as artificial neural networks (ANN) are used as critical components of the study.

Hyper parameter tuning

The study examined several distinct facets of machine learning algorithms, with the primary goal of improving their performance through the difficult process of hyperparameter tweaking. This procedure is vital for fine-tuning the critical settings of these models to attain the best prediction accuracy. A parameter grid is methodically created to explore this hyperparameter field systematically. A variety of hyperparameters are included, including the number of estimators (trees), the maximum depth of these trees, and the minimum sample required for node splitting.

GridSearchCV (Pirjatullah et al., 2021), (Gill & Gupta, 2023) is a strategy that rigorously investigates numerous hyperparameter arrangements to discover the optimum set for a machine learning model. It comprises generating a set of hyperparameter values as a grid, testing all possible combinations, evaluating model performance, and picking the best one, therefore assisting in model optimization. Figure 6 depicts how GridSearchCV hyperparameter tuning works. The parameter grid and GridSearchCV best outcome for all the models are tabulated in Table 2.

Figure 6 A depiction of how Grid Search CV functions.

Table 2 Employing various parameters in every algorithm to avoid overfitting.

Classifier	HyperParameters	Possible values	GridSearchCV outcome	
Random forest	Number of estimators	50, 100, 200	50	
	Maximum depth	None, 10, 20	None	
	Minimum samples split	2, 5, 10	2	
	Minimum samples leaf	1, 2, 4	1	
	Maximum features	sqrt, log2	sqrt	
Extra trees	Number of estimators	50, 100, 200	50	
	Maximum depth	None, 10, 20	None	
	Minimum samples split	2, 5, 10	2	
	Minimum samples leaf	1, 2, 4	1	
	Maximum features	sqrt, log2	sqrt	
LGBM	Minimum child samples	5, 10, 20	5	
	Maximum depth	5, 10, 15	5	
	Number of estimators	50, 100, 200	100	
	Learning rate	0.001, 0.01, 0.1	0.1	
Decision tree	Minimum samples leaf	1, 2, 4	1	
	Maximum depth	None, 10, 20	None	
	Minimum samples split	2, 5, 10	5	
	Maximum features	auto, sqrt, log2	sqrt	
Logistic regression	C	0.001, 0.01, 0.1, 1, 10, 100	10	
	Penalty	L2	L2	
ANN	Hidden layer sizes	64,32,128	64	
	Alpha	0.0001, 0.001, 0.01	0.0001	
	Learning rate	0.001, 0.01, 0.1	0.01	
	Activation	ReLU, TanH	ReLU	
	Solver	Adam, SGD	SGD	

ANN

An artificial neural network (ANN) with an input layer, two hidden layers incorporating the rectified linear unit (ReLU) activation function, and an output layer utilizing the sigmoid activation function for binary predictions is used in the context of binary classification for chronic renal disease (CRD). The network design is subjected to a grid search for hyperparameter tuning, which involves experimenting with various combinations of hidden layer sizes, activation functions (ReLU, tanh), solvers (Adam, SGD), regularisation strengths (alpha), and starting learning rates.

The Adam optimizer is used during model training to repeatedly change the model parameters. The grid search tests multiple hyperparameter combinations systematically using 5-fold cross-validation, to discover the configuration with the highest predictive performance.

The grid search discovers the most effective design by examining several sets of hidden layer sizes, including (64,), (32,), (128, 64), and (64, 32, 16). The chosen model obtains a outstanding test accuracy of 99.0%, suggesting its capacity to properly detect and forecast the existence of CRD.

Extra tree classifier

The Extremely Randomized Trees also known as extra trees classifier (Geurts, Ernst & Wehenkel, 2006), which is highly skilled in group learning, is applied carefully. The GridSearchCV technique (Pirjatullah et al., 2021) is used with great care in this implementation, methodically exploring a large range of hyperparameter combinations. To prevent overfitting during this exhaustive process, deliberate adjustments are made to several critical parameters. These adjustments include determining the minimum sample size required to initiate splitting in a tree node, limiting the maximum depth of individual trees, and specifying the total number of trees within the classifier’s “forest”. Additionally, balanced data is consistently utilized throughout our analysis.

The principal objective of this systematic investigation is to identify the ideal hyperparameter combination that would optimize the model and allow it to ascertain the presence of CRD, with the highest possible degree of accuracy. The results of this thorough investigation are extremely impressive. An astounding accuracy of 99.04% on the test dataset is achieved by using the grid search to identify the ideal hyperparameter configuration. With its ability to capture the nuances of the dataset and produce incredibly accurate predictions, this outstanding accomplishment demonstrates the extra trees classifier’s potential as a potent tool in CRD prediction.

Logistic regression

Logistic regression (LR) is a fundamental linear modelling technique in predictive analytics. Critical variables such as solver, penalty, and regularisation strength were thoroughly examined using the GridSearchCV approach.

The solver parameter defines the optimisation strategy used in logistic regression, whilst the penalty determines the type of regularisation used. The emphasis was mostly on L2 regularisation, a variation of the elastic net technique that supplements the algorithm’s loss function to prevent overfitting and encourage a well-generalized model.

The LR model obtained 99.07% accuracy on the experimental dataset by carefully tuning hyperparameters. This outstanding accuracy puts LR on par with other top-performing models, reaffirming its efficacy in predicting CRD.

Furthermore, the stability of LR’s higher performance solidifies its position as the preferable model for CRD prediction. Its consistent supply of accurate findings, which outperform competing models, demonstrates its applicability for real-world applications. As a result, LR was chosen as the foundation for building the entire programme, offering trustworthy predictions that are critical for influencing clinical decision-making and improving patient outcomes. The mathematical intuition is as follows: (4) Lossw=−∑i=1Nai logpai∣xi,w+1−ailog1−pai∣xi,w+λ∥w∥2

Loss(w) is the logistic regression loss function with L2 regularization.

• N is the overall amount of observations.

• ai is the actual binary label for data point i.

• xi is the data point (i) feature vector.

• w is the vector of model parameters (weights).

• p(yi − xi, w) is the probability of the positive class predicted by the model for data point i.

• lambda(λ) is the regularization strength, determining the influence of the L2 penalty on the weight of the model.

• ∥w∥2 reflects the weight vector w’s L2 norm (Euclidean norm).

L2 term λ||w||2 has been added to the standard logistic regression loss function to prevent overfitting. It penalizes large values of the model’s weights (w), effectively encouraging smaller, more balanced weight values. This helps in producing a more robust and generalized logistic regression model.

__________________________________________________________________________________________ Algorithm 2 ML Model Training using GridSearchCV Hyperparameter Tuning   1:  Input:    Sampled  data  Xtrain_resampled,   Xtest_resampled,   ytrain_resampled,      ytest_resampled = resample_data(Xtrain,ytrain)   2:  Create a scaling pipeline:   3:  Xtrain_scaled,Xtest_scaled = scale_data(Xtrain_resampled,Xtest_resampled)   4:  Define hyperparameter grid for grid search   5:  Initialize best accuracy variable, best_test_acc = 0   6:  Iterate over hyperparameter grid   7:  for each hyperparameters in hyperparameter_grid do  8:       Build ML model with specified hyperparameters   9:       model = build_ml_model(hyperparameters) 10:       Train the model 11:       model.fit(Xtrain_scaled, ytrain_resampled) 12:       Evaluate on the test set 13:       test_acc = model.evaluate(Xtest_scaled, ytest_resampled) 14:       Update best accuracy and best model if the current model is better 15:       if test_acc > best_test_acc then 16:            best_test_acc ← test_acc 17:            best_model ← model 18:       end if 19:  end for 20:  Display results 21:  display_results(best_model, best_test_accuracy)_____________________________________

Random forest

In this research, we used the RandomForest classifier because of its shown success in creating highly accurate predictions by using the collective knowledge of decision trees. This methodology builds on our previous usage of the extra trees classifier, emphasizing the importance of ensemble learning approaches in solving complicated challenges like predicting CRD. To improve the model’s prediction performance and also to remove overfitting, we tested several parameters in our RandomForest implementation. Bootstrap, Minimum Samples Split, Minimum Samples Leaf, Maximum Features, Maximum Depth, and number of estimators are among the factors used. Each hyperparameter influenced many parts of the model’s behavior, ranging from resampling strategies to tree depth and branching. The comprehensive hyperparameter tuning efforts attempted to establish the optimal configuration for fully realizing the model’s potential in CRD prediction. Notably, these efforts are rewarded, as the RandomForest classifier attained an exceptional accuracy value of 99.05% on the test dataset. This outstanding accomplishment illustrates the model’s exceptional capacity to generate extremely accurate forecasts and solidifies its expertise in the field of CRD prediction.

LGBM

The study used the LightGBM classifier, a gradient-based learning algorithm known for its effectiveness in predictive modelling (Ke et al., 2017). The model obtained outstanding predicted accuracy of 98.15% on the test dataset by meticulously optimising crucial hyperparameters such as learning rate, maximum tree depth, and boosting rounds using GridSearchCV. This achievement highlights LightGBM’s robustness in CRD prediction, proving its potential to produce exact prognostications. Furthermore, LightGBM’s approach to minimising an objective function that includes both prediction errors and model complexity guarantees a balanced and reliable framework for CRD prediction problems, making a substantial contribution to the growth of predictive analytics in healthcare. Indeed, the description presented provides a concise synopsis of the LightGBM algorithm’s fundamental mechanisms. At its core, LightGBM seeks to minimise an objective function that consists of two key components: the loss term and the regularisation term.

1. Loss term: This component assesses the prediction error for each data point and works to reduce these mistakes collectively. The loss term helps to refine model predictions by efficiently capturing the differences between projected and actual values, thus boosting overall accuracy.

2. Regularisation term: The regularisation term is designed to reduce model complexity and prevent overfitting. This term ensures that the model generalises successfully to unknown data by setting constraints on its structure and parameters, hence increasing its dependability and resilience. (5) Objective= ∑Ly,y ˆ+ ∑Ωg

where:

L(y, ŷ) is the loss term measuring prediction errors.

Ω(g) is the regularization term penalizing complex decision trees.

LightGBM employs gradient descent to reduce this function and increase model accuracy. To summarise, it combines effective approaches with gradient boosting to obtain accurate prediction.

Decision tree

In the decision tree classifier, meticulous hyperparameter tuning using GridSearchCV was employed on resampled data to enhance model performance. While the decision tree exhibited notable efficacy, achieving an accuracy of 97.22%, its performance slightly trailed behind other advanced models utilized in our study. This detailed investigation highlights the complexities of CRD prediction, needing a careful examination of numerous algorithms and configurations to determine their respective strengths and limits. Despite its slightly poorer performance, the decision tree’s interpretability and ability to capture subtle relationships within the data provided vital insights into our overall goal of developing a robust CRD prediction model.

Results and Analysis

The analysis thoroughly examined numerous categorization systems to determine the most efficient strategy for identifying CRD. The study attempted to give a full evaluation by using approaches such as LightGBM (LGBM), LR, RF, extra trees classifier, DT and ANN.

Validation methods are strictly followed, which included dividing the dataset into training and testing sets. Notably, the study used sophisticated strategies to overcome class imbalance, such as the SMOTE, and hyperparameter adjustment to optimize algorithm performance. The assessment measures employed, such as accuracy, precision, recall, and F1-score, offered a thorough grasp of each algorithm’s diagnostic competency in detecting people with CRD. Table 3 highlights the differences in accuracies obtained by each classifier before and after using GridsearchCV for hyperparameter tuning. Table 4 shows the comparison of all the classifiers based on essential evaluation metrics such as precision, specificity, recall, accuracy, f1-score and AUC, which provides the valuable insights into the performance of each classifier.

Table 3 Accuracies before and after hyperparameter tuning.

Classifier	Before hyperparameter tuning	After hyperparameter tuning	
Random forest	97.22	99.07	
Extra trees	96.3	99.07	
LGBM	96.3	98.00	
Decision tree	95.37	97.03	
Logistic regression	97.22	99.07	
ANN	96.03	99.07	

Table 4 Evaluation metrics for various classifiers.

Classifier	TN	FP	FN	TP	Precision (%)	Recall (%)	Accuracy (%)	F1-Score (%)	Specificity (%)	AUC (%)	
Random forest	56	1	0	51	98.08	100	99.07	99.03	98.25	98.95	
Extra trees	57	0	1	50	100	98.04	99.07	99.01	100	99.40	
LGBM	57	0	2	49	100	96.08	98.15	98.00	100	98.80	
Decision tree	56	1	2	49	98.00	96.08	97.22	97.03	98.25	97.00	
Logistic regression	57	0	1	50	100	98.04	99.07	99.01	100	99.40	
ANN	57	0	1	50	100	98.04	99.07	99.01	100	99.40	

These findings highlight the study’s focus on methodological rigor, utilizing cutting-edge techniques to achieve excellent predictive performance in CRD diagnosis. Figures 7 and 8 serve as visual aids, depicting the outcomes of each algorithm’s review procedure in a straightforward manner.

Figure 7 Comparison of performance metrics of all classifiers.

Figure 8 Testing vs training accuracy of all classifiers.

Evaluation metrics

Recall

True positive rate (TPR) and sensitivity are other names for recall (R). By dividing the total number of true positives by the number of true positive and false negative predictions, it is calculated. It is crucial to recognize the strong correlation between the number of positive samples detected and the recall measure. (6) R=True PositivesTrue Positives + False Negatives.

An ML model’s recall is termed poor when the total amount of true positives (TP) and false negatives (FN) in the denominator exceeds the TP in the numerator. When the TP in the numerator surpasses the TP+FN in the denominator, the model’s recall is deemed high. This implies that the number of FNs should be low for having a higher recall.

TN: Instances correctly classified as negative by the model. TP: Instances correctly classified as positive by the model. FN: Instances incorrectly classified as negative by the model. False Positives (FP): Instances incorrectly classified as positive by the model. If an instance is classified as negative, it means no kidney disease is identified from the provided attribute information. If classified as positive, it means kidney disease is identified.

Specificity

True negative rate (TNR), another name for specificity, is obtained by dividing the number of true negative predictions by the sum of true negatives and false positives. The number of FPs should be low for having a higher specificity. (7) S=True NegativesTrue Negatives + False Positives.

Precision

Precision (P) is characterized as the ratio of actual positive samples rightly identified among all samples classified as positive by the model. (8) P=True PositivesTrue Positive + False Positives.

The precision of an ML model will be high when the value of TP (numerator) ≥ TP+FP (denominator) and it will be low if TP+FP (denominator) ≥ TP (numerator). This implies that the number of FPs should be low for having a higher precision.

F1-score

The F1-score is an evaluation statistic that is used to assess the efficacy of the ML model. It utilizes both precision and recall, making it appropriate for datasets with imbalances. (9) F1−score=2⋅P * RP + R.

The F1-score, which ranges from 0 to 1, represents the performance of the model, with 0 being the worst situation and 1 indicating the most favorable case. By offering a weighted average of precision and recall, this score compensates for both erroneous positives and false negatives.

The F1-score will be high if both precision and recall exhibit elevated values. In contrast, if both precision and recall are poor, the F1-score will be minimal. When one of these indicators is low and the other is high, the F1-score will be in the medium range.

Accuracy

In the field of AI and ML, we evaluate a model’s performance using accuracy (A), which is commonly stated as a percentage. It measures the proportion of correct predictions to total predictions, providing a measure of the model’s accuracy. (10) A=Number of Correct PredictionsTotal Number of Predictions.

Discussion

From the results, we can clearly see how using GridsearchCV for hyperparameter tuning leveraged the performance of all the classifiers. Out of all the classifiers we considered, random forest, extra trees, logistic regression with L2 penalty, and ANN provided the highest accuracy of 99.07%. However, due to the stochastic nature of ML, the results vary each time. Therefore, we considered the average metrics, which still performed well. Upon close monitoring during result evaluation, logistic regression demonstrated greater consistency compared to the other classifiers. The phenomenon of logistic regression with L2 penalty consistently performing well can be attributed to its ability to effectively handle feature regularization and prevent overfitting. Along with accuracy, we have also considered other evaluation metrics, which show the potential of each of these classifiers. Extra trees, LGBM, logistic regression with L2 penalty and ANN showcased a 100% precision, while random forest recorded its 100% recall.

These highly accurate results motivated us to build a real-time end-to-end application for CRD prediction using ML algorithms. Figure 9 depicts the prediction of “No kidney disease” with class label 0 for the user data.

Figure 9 Sample CRD negative prediction from kidney disease predictor.

Interpreting ML models using XAI

While the models achieved remarkable performance, their inner workings are still opaque, leaving users with little understanding of how predictions are made. This black-box nature of ML models has long been a challenge in deploying them in critical applications. Devam, Het Naik & Patel (2023) explains how lack of transparency raises concerns about trust, accountability, and ethical implications, particularly in high-stakes domain like healthcare. Explainable AI (XAI) techniques have emerged as a solution to address these challenges by providing insights into the decision-making process of ML models. In our work, we have considered Local Interpretable Model-agnostic Explanations and SHapley Additive exPlanations, which are two prominent XAI techniques that offer interpretability for complex models.

Figure 10 illustrates the working of XAI, the explainer model m(y) produced by XAI techniques such as SHAP or LIME tends to mimic the behavior of the original model n(y). This phenomenon occurs because the explainer model is trained to approximate the behavior of the original model within a local neighborhood of the instance being explained.

Figure 10 Working of XAI.

SHAP (Lundberg & Lee, 2017) provides globally consistent explanations by computing Shapley values, which measure the contribution of each feature to the prediction across all possible combinations. This allows users to understand the relative significance of various features and their impact on the model’s output. Conversely, LIME (Ribeiro, Singh & Guestrin, 2018) focuses on creating local explanations by approximating the behavior of the ML model around specific instances. It does this by training a simpler, interpretable model on perturbed instances around the instance of interest and using the model’s predictions to explain the original model’s decision locally. Together, SHAP and LIME empower users to gain deeper insights into ML models’ decisions, establishing trust, transparency, and accountability in AI-driven systems .

SHapley additive ExPlanations

SHAP is a strong method that provides explanations for the output of any machine learning model, regardless of internal complexity. In SHAP, the mean forecast (base value) of the model is established, and each feature’s relative contribution to the target’s divergence from the base is determined. It is able to provide both global and local explanations.

First, let’s examine the global explanations using the SHAP summary plot, as depicted in Fig. 11. The SHAP summary plot illustrates the impact of various features on the model’s predictions for chronic kidney disease. The y-axis represents the features such as “haemoglobin”, “albumin”, and “blood_urea”, with the longest bars indicating the highest influence. The x-axis indicates the mean of the absolute SHAP value, showing the average impact of each feature on the model’s output. The adjacent red and blue bars represent the contribution towards predicting CRD positive (Class 1, red) or CRD negative (Class 0, blue). The length of each bar shows the overall impact of the feature on the model’s outcomes, but it does not indicate whether higher or lower values lead to a specific prediction. This plot helps identify which features are most influential in the model’s decision-making process, enhancing the interpretability of the predictive model.

Figure 11 SHAP summary plot.

Figure 12 SHAP force plot for CRD negative prediction.

Now for local explanations, we consider the SHAP Force Plot, which can be seen in Fig. 12. The horizontal line on the force plot spans values from −20.79 to 29.21, with a baseline value marked at 0. The labels ‘higher’ and ‘lower’ denote the effect of features on the prediction. Features can either increase (push towards ‘higher’) or decrease (push towards ‘lower’) the prediction. This force plot pertains to a specific instance, shedding light on how particular features, such as specific gravity, hypertension, packed cell volume, appetite, albumin, and red blood cell count, influence the model’s predictions. Only the top six features’ contributions are considered, and for this instance, all these features collectively contribute to lowering the prediction. The predicted label for this instance is 0, and the force plot elucidates why the model assigned this label.

The value f(x) in the force plot represents the predicted log-odds of the outcome for the specific instance. This log-odds value is transformed into a probability p(x) using the logistic function, which is defined as p(x) = 11+e−fx. In this context, when f(x) = −12.29 then p(x) = 2.9∗10−6 which is extremely close to 0. Since this probability is well below 0.5, it is classified as the negative class (0).

In the Fig. 13, we observe that the length of the red side, indicating positive contributions, is longer than the blue side, representing negative contributions. This suggests a positive outcome. The features driving this outcome include albumin, blood urea, blood pressure, and red blood cells, which exert positive contributions. Conversely, haemoglobin, specific gravity, and packed cells attempt to lower the outcome but do not contribute significantly enough. Therefore, the overall effect leans towards a positive outcome.

Figure 13 SHAP force plot for CRD positive prediction.

Similarly, here f(x) = 0.88 then p(x) = 0.74, comparing this probability to the threshold of 0.5, it is classified as the positive class (1).

Local interpretable model-agnostic explanations

LIME is a method for explaining machine learning models’ predictions on individual instances by approximating the decision boundary locally around the instance of interest and generating interpretable models. A LIME plot is a visual representation generated by the LIME technique to explain the predictions of a machine learning model for an individual instance.

In Fig. 14, we can see a LIME plot explaining the prediction made for a CRD positive instance. The prediction probabilities on the left indicate the model’s confidence in predicting the class for the given instance. The dominant 0.92 probability orange bar indicates a 92% confidence that the instance is CRD positive, while the 0.08 probability blue bar indicates only an 8% confidence that the instance is CRD negative. Hence, the instance is identified as CRD positive. The plot includes the top six features along with their values and contributions. For example, specific_gravity has a value of −0.40 in this instance, and its contribution is 0.13 towards CRD positive, while blood_glucose_random has a value of −0.72 and its contribution is only 0.04 towards CRD negative. Here, value means the feature value that a feature has for that instance, which is obtained from the dataset. For example, peda_edema is 1.96 . In the bar graph representation, peda_edema > −0.49 with a 0.04 contribution indicates that peda_edema having any value greater than −0.49 will contribute to positive outcome, and here with 1.96 value, the contribution of peda_edema is 0.04 towards the positive outcome. Features such as specific gravity, packed cell volume, haemoglobin, peda edema and hypertension exhibit larger positive contributions, whereas blood glucose random shows smaller negative contribution. As a result, the prediction leans towards a positive outcome. Our model correctly predicted this instance as CRD positive, and the LIME plot provides a visual representation of the features that influenced the machine’s decision-making.

Figure 14 LIME plot for CRD positive prediction.

Similarly, in Fig. 15, a CRD negative prediction is depicted, characterized by higher contributions from features such as packed cell volume, specific gravity, hypertension, serum creatinine and diabetes mellitus, compared to the lower contribution of red blood cells. As a result, the prediction is negative.

Figure 15 LIME plot for CRD negative prediction.

Conclusion

Our work concludes by demonstrating the effectiveness of ML methods in early prediction of CRD. By leveraging a diverse range of ML algorithms, we have showcased the potential of predictive modelling in supporting clinical decision-making processes. Through rigorous evaluations and the integration of XAI techniques such as LIME and SHAP, we have offered insightful information about the variables influencing model predictions, enhancing interpretability and trust among medical practitioners. Our approach addressed various challenges, including missing values, irregular data ranges, and class imbalance, through strategic methods such as missing value imputation, data scaling, and SMOTE, respectively. Furthermore, hyperparameter optimization using GridSearchCV enabled us to fine-tune model performance and achieve remarkable accuracy rates, with random forest, extra trees classifier, logistic regression with L2 penalty and ANN emerging as the top-performing models, each attaining a 99.07% accuracy rate. After close monitoring and multiple executions, we have determined that logistic regression with L2 penalty consistently outperformed the other classifiers, establishing itself as the superior model for predicting CRD. Our findings underscore the significance of early CRD prediction and the potential of XAI-ML approaches in proactive disease management. A fully functional app has also been developed to facilitate real-time CRD prediction. Looking ahead, research focused on building secure ML and XAI models can significantly enhance the applicability of these methods in real-world healthcare settings, thereby improving global healthcare outcomes.

Supplemental Information

Supplemental Information 1 Chronic Kidney Disease Dataset

Additional Information and Declarations

Competing Interests

Author Contributions

Data Availability

The authors declare there are no competing interests.

Sanjana Singamsetty conceived and designed the experiments, performed the experiments, analyzed the data, performed the computation work, prepared figures and/or tables, authored or reviewed drafts of the article, and approved the final draft.

Swetha Ghanta conceived and designed the experiments, performed the experiments, analyzed the data, performed the computation work, prepared figures and/or tables, authored or reviewed drafts of the article, and approved the final draft.

Sujit Biswas conceived and designed the experiments, performed the experiments, analyzed the data, performed the computation work, prepared figures and/or tables, authored or reviewed drafts of the article, and approved the final draft.

Ashok Pradhan conceived and designed the experiments, performed the experiments, analyzed the data, performed the computation work, prepared figures and/or tables, authored or reviewed drafts of the article, and approved the final draft.

The following information was supplied regarding data availability:

The code is available at GitHub and Zenodo:

- https://github.com/sanjana-singamsetty/kidney_disease_prediction.

- Singamsetty, S. (2024). Enhancing Machine Learning-based Forecasting of Chronic Renal Disease with Explainable AI (Version 1). Zenodo. https://doi.org/10.5281/zenodo.13334915.

The Chronic Kidney Disease dataset is available in the Supplemental File and the UCI Machine Learning Repository: Rubini,L., Soundarapandian,P., and Eswaran,P.. (2015). Chronic Kidney Disease. UCI Machine Learning Repository. https://doi.org/10.24432/C5G020.

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
