# Peer review of "Enhancing machine learning-based forecasting of chronic renal disease with explainable AI"

_PeerJ Computer Science, doi:10.7717/peerj-cs.2291_

## Round 0.1 · original submission · Major Revisions

The reviewers have substantial concerns about this manuscript. The authors should provide point-to-point responses to address all the concerns and provide a revised manuscript with the revised parts being marked via tracked changes.

·

Basic reporting

This work provides an end-to-end predictive model for CRD diagnosis to enhance patient outcomes. This approach achieved a high predictive accuracy, especially for logistic regression with L2 penalty. The context is clear and unabiguous with appropriate English language.

Experimental design

1. The author listed different ML and DL approaches from previously published works in Table 2. It seems that most of those approaches have very high accuracy (> 97%). Please address the rationale of comparing these different approaches. Which approach is more preferred? How the authors benefit from these approaches?
2. How is the specificity of this ML method in predicting CRD? Which factors/parameters have the great impact on positive/negative prediction?

Validity of the findings

Please include figure legends in each figure.

Reviewer 2 ·

Basic reporting

In this manuscript, Singamsetty and colleagues have studied six different machine learning-based models to classify chronic renal disease. They improved the model performance using hyperparamter tuning and further investigated feature importance. While the paper provides a good analysis framework for disease prediction, several areas could benefit from further attention:
1. Abstract, please be more specific about the use case of the predictive model – binary disease classification.
2. Kindly consider having a fluent English speaker proofread the manuscript for accuracy and clarity. For instance, certain phrases such as page 2 line 68, page 4 line113, page 7 line 119 and page18 line 395 could be revised for better clarity.
3. Page 4 line 111, please be more specific about the “attribute correlation”. How is this calculated? by Pearson correlation?
4. Please be more specific about the features included in the web app and provide a link to the app.
5. Since this is a binary classification problem, there is no need to discuss the stages of CRD in such detail.
6. Please improve the resolution of figures, such as fig. 5, fig. 7 and fig. 11.
7. Several formulas, such as 3-5 and 8-11 are very basic and there is no need to define
8. Fig. 6, the “validation evaluation results” part is confusing. The model applied to the testing data should not be used for hyperparameter tuning.
9. Please provide figure legends across all the figures. Some of the main text should be moved to the figure legend, such as line 395-401 and line 402-406.
10. Feature importance can also be inferred with the tree models and logistic regression. How are those features ranked comparable to SHARP and LIME? Are there any important features shared, and why?
11. Please provide a valid github link for all the analyses.

Experimental design

no comment

Validity of the findings

no comment

Additional comments

no comment

Reviewer 3 ·

Basic reporting

Figures and tables are provided, but the explanations accompanying them are insufficient. Please see below comments for more details.

Experimental design

Six machine learning algorithms were utilized to conduct the analysis. However, the entire experimental design lacks novelty. Despite achieving high accuracy, overfitting is suspected. Please see below comments for more details.

Validity of the findings

The statistical summary is not provided for the original dataset, which impacts the reader's understanding of the article. The article does not seem to mention the number of records in the original dataset, which might not be sufficient to yield a comprehensive result. Predicting a binary variable appears to be overly simplistic. Please see below comments for more details.

Additional comments

1. Line 68: The sentence "After rigorous tests using…" is incomplete.
2. Line 83: Please spell out the full name of "DL" since it's the first time it is mentioned.
3. Lines 108–115: The description of the dataset is insufficient. Please provide more information, such as a statistical summary including the value for each attribute, the count of each attribute's records, and the count of CRD positive/negative for each value. What is the total count (N) of the dataset? In lines 110-111, is 400 the count? What does the class attribute mean? Please provide further explanation. In line 114, what does 158 mean? Does it mean 158 records out of all records have missing values? Or are there only 158 fields in the whole dataset with missing values? Please provide further explanation.
4. Lines 174–185: One of the possible effects of using SMOTE is overfitting. Is an accuracy of 99.07% indicative of overfitting? Please add the number of records used in the model before and after SMOTE.
5. Line 190: The reviewer suggests adding evaluation results before/after parameter tuning or for each potential parameter combination to show the difference and improvement of results using GridSearchCV. Taking accuracy as an example, before parameter tuning, the accuracy is 92%, and after finding the best parameter, the accuracy is 99%. These numbers will intuitively show the effect of parameter tuning.
6. Line 306: Again, please provide the number of records in the dataset. How many for the training set and how many for the testing set?
7. Table 4: What are "TN, FP, FN, TP"? They are not explained in the manuscript.
8. Figure 8: This figure doesn't provide much meaningful interpretation. The height of each group of bars is similar and hard to differentiate with the naked eye. The values are not shown in the bar chart. If not presenting the same evaluation parameter of each classifier together (for example, putting the precision of the 6 classifiers together), it's really hard to discern the difference and make comparisons.
9. Lines 357–358: Do the four classifiers give the exact same accuracy?
10. Lines 395–402: The meaning of the illustration in this paragraph is obvious. But the description doesn't explain Figure 11 well. Please further explain.
11. Lines 423–433: The description of Figures 14 and 15 is insufficient. In lines 427–428, what does "orange bar represents a positive prediction" mean? Why would there be a negative value when predicting a positive value? What do the numbers in the table mean? What's their significance? In the bar chart, what do the numbers on the bars mean (e.g., in Fig.14, 0.12 of specific gravity)? What does the number of the feature threshold mean (e.g., in Fig.14, <=-0.42 of specific gravity)? What do the positive and negative signs in these values mean? What do the numbers in the legend represent in the chart/table? Why is there no color for the legend in Fig. 15? Please further address the above issues.
12. Lines 434–452: The results show certain features are important for CRD prediction. Does this result align with research experience in the medical field? Is there any validation with practical medical research or experience?

---

## Round 0.2 · accepted · Accept

Reviewers are satisfied with the revisions, and I concur to recommend accepting this manuscript.

Reviewer 2 ·

Basic reporting

I appreciate the author's response to my questions, which addresses all my concerns.

Experimental design

I appreciate the author's response to my questions, which addresses all my concerns.

Validity of the findings

I appreciate the author's response to my questions, which addresses all my concerns.

Reviewer 3 ·

Basic reporting

no comment

Experimental design

no comment

Validity of the findings

no comment

Additional comments

I appreciate the author's thorough analysis and response to the previous review comments. The revised version has effectively resolved the issues highlighted earlier. The language is now fluent, the explanations are clear, and figure legends have been added along with more comprehensive explanations. Additionally, the structure and presentation have significantly improved, making the article easier to follow. The enhanced clarity and detail in this version suggest that it is now ready for acceptance from my opinion.